



# Extremes of surface snow grains change in East Antarctica and their relationship with meteorological conditions

Claudio Stefanini[1,2], Giovanni Macelloni[2], Marion Leduc-Leballeur[2], Vincent Favier[3], Benjamin Pohl[4], and Ghislain Picard[3]

[1]Dipartimento di Scienze Ambientali, Informatica e Statistica, DAIS, Ca' Foscari University of Venice, Mestre-Venezia, Italy
[2]Institute of Applied Physics "Nello Carrara" – National Council of Research, 50019 Sesto Fiorentino, Italy
[3]Université Grenoble Alpes, CNRS, IRD, Grenoble INP, IGE, 38000 Grenoble, France
[4]Biogéosciences, UMR6282 CNRS / Université de Bourgogne, Dijon, France

**Correspondence:** Stefanini (claudio.stefanini@unive.it)

**Abstract.** This study explores the grain size seasonal variations on the East Antarctic Plateau, where dry metamorphism occurs, by using microwave radiometer observations from 2000 to 2022. Local meteorological conditions and large scale atmospheric phenomena have been considered in order to explain some peculiar changes of the snow grains. We find that the highest ice divide is the region with the largest grain size in the summer, mainly because the wind speed is low. Moreover, some extreme grain size values with respect to the average (over $+3\sigma$) were identified. In these cases, the ERA5 reanalysis revealed a high pressure blocking/ridge situation in the proximity of the onsets of the summer increase of the grain size, conveying the relatively warm and moist air coming from the mid latitudes, often associated with atmospheric rivers. If weak wind and low temperature conditions occur during the following weeks, dry snow metamorphism is facilitated, leading to grain growth. This determines anomalous high maximums of the snow grain size at the end of summer. These phenomena confirm the importance of moisture intrusion events in the East Antarctica and their impact on the physical properties of the ice sheet surface, with a co-occurrence of atmospheric rivers and seasonal changes of the grain size significant over 95%.

## 1 Introduction

The seasonal variations of the grain size observed by spaceborne microwave radiometer in Antarctica shows a large summer increase, which is mainly driven by temperature (Picard et al., 2012). The increase in temperature during austral summer leads to coarsening of snow grains due to grain-to-grain transport of water vapour (Colbeck, 1982, 1993; Sturm and Benson, 1997; Flanner and Zender, 2006; Town et al., 2008), while winter precipitations and wind-transported snow accumulate small grains over the surface (Domine et al., 2007). On the East Antarctic Plateau the mean annual precipitation is low, less than $100\,\mathrm{kg\,m^{-2}}$ in the highest part of the ice sheet, due to its insulation from the rest of the hemisphere (Palerme et al., 2014). However, some relatively warm and moist oceanic air masses occasionally travel far inland providing higher than usual snow accumulations (Turner et al., 2019). These intrusions could be linked to atmospheric rivers known to transport warm and moist air from the tropics to the polar latitudes (e.g. Wille et al., 2021).



Understanding and modelling the physical changes in snow properties, and in particular the grain size, is important to predict the changes in snowpack albedo (Grenfell et al., 1994; Domine et al., 2006), and hence of surface energy balance at high latitudes. Moreover, it provides information on the firn properties, useful to retrieve the ice sheet mass balance by space-borne altimetry (Keenan et al., 2021; Kingslake et al., 2022). On the Antarctic Plateau, in-situ snow structure observations are rare and the meteorological measurements, needed to interpret the causes of changes in the snowpack, are sparse. This is the reason for the constant need of supplementary data coming, for example, from satellite observations, able to provide continuous information on snow surface characteristics at continental scale. Previous studies investigated the snow grain size retrieved from remote sensing observations. In Jin et al. (2008) near infrared and visible satellite observations were exploited to retrieve the grain size in the top 1 cm during clear sky conditions and observed widespread summer increase of grain size. Picard et al. (2012) combined high frequency passive microwave observations to implement an indicator of grain size in the uppermost ∼10 cm of the snowpack, and compared it to observations at Dome C and simulation results obtained by modelling water vapour transport near the surface. They observed inter-annual variability in the summer grain growth and linked them to precipitation variability. Also Casado et al. (2021) focused on Dome C, deepening the link between water isotopic signature and surface snow metamorphism observing changes in this grain size indicator for two contrasted summers (2014 and 2015). Hence, past studies were limited to clear sky conditions or to a particular location (Dome C). Our work, by means of high frequency microwave observations explores the interaction between surface snow grain size and meteorological conditions, on a continental scale and on a seasonal time frame.

This work focuses on extremes of high grain size. The investigated events were recorded during summer between 2000 and 2022, close to the highest elevation points of the East Antarctica Plateau, along the ice divide. Using both passive microwave observations and atmospheric reanalysis, we investigate the physical processes involved in these events and their link with the local and regional atmospheric conditions.

The paper is organised as follows. Section 2 and Section 3 present the satellite and meteorological datasets and statistical tools. Section 4 shows the results by first describing the common features of the grain size summer increases and their relationship with local meteorological conditions (Section 4.2); second, focusing the analysis to the four most extreme grain size increases (Section 4.2) and lastly, connecting the changes of the grain size with large scale meteorological conditions (Section 4.3). In Section 5 we discuss the results and Section 6 draws conclusions.

## 2 Materials

### 2.1 Passive microwave observations

We use the Advanced Microwave Sounding Unit B (AMSU-B), which is a 5-channel microwave radiometer operating since 1998 on-board of NOAA-15 to NOAA-19 and METOP satellites. Swath data from the sensors NOAA-15 to NOAA-19 have been downloaded from the National Center for Environmental Information (https://www.avl.class.noaa.gov/) and after selecting incidence angles in the range 0–25°, processed to obtain daily mean from January 2000 to May 2022 in Southern polar stereographic projection (EPSG: 3976) at a resolution of 25 km. Brightness temperatures observed at 89 and 150 GHz are



then used to compute the snow *Grain Size Index (GSI)* defined as $1 - T_B(150\,\mathrm{GHz})/T_B(89\,\mathrm{GHz})$ by Picard et al. (2012). They focused their study on Dome C, and provided an approximate correspondence between the index and millimeters: 0.00 of GSI is $\sim 0.025\,\mathrm{mm}$ and 0.20 is $\sim 0.150\,\mathrm{mm}$ (for the grains in the first $7\,\mathrm{cm}$ of the snowpack). Because of the particular and high sensitivity of the microwaves to liquid water, GSI is only effective in regions where snow melt never occurs, such as the East Antarctic Plateau (Picard et al., 2007).

## 2.2 Meteorological data

### 2.2.1 ERA5 reanalysis

For this study, we use skin temperature, $2\,\mathrm{m}$ temperature, $10\,\mathrm{m}$ wind speed, snowfall, surface pressure, cloud cover and surface downward long–wave and short–wave radiation flux over Antarctica from the ERA reanalysis produced by the European Centre for Medium-Range Weather Forecasts (ECMWF, Hersbach et al., 2018a, b). Data are provided over a regular latitude-longitude grid of $0.25°$, and we reprojected them over the Southern polar stereographic projection. We used daily mean from 2000 to 2022.

### 2.2.2 Atmospheric river catalogs

The atmospheric river presence is obtained from the two catalogs built by Wille et al. (2021). The detection is based on the integrated water vapour (IWV) and the meridional component of the integrated vapour transport (vIVT), both from 3 hourly data of the Modern-Era Retrospective analysis for Research and Applications, Version 2 (MERRA-2) reanalysis (Gelaro et al., 2017). The datasets provide a boolean indicator of the atmospheric river presence for each grid cell worldwide with a latitude–longitude resolution of $0.50° \times 0.625°$. Previous work revealed weak sensitivity to the input meteorological dataset used, between MERRA-2 (used for the tier-2 exercise of the ARTMIP project: Collow et al. (2022)) and ERA5, especially over East Antarctica (Pohl et al., 2021; Wille et al., 2021).

### 2.2.3 Antarctic Oscillation index

The Antarctic Oscillation index (AAO, also known as the Southern Annular Mode, SAM) (Thompson and Wallace, 2000) is provided by the National Oceanic and Atmospheric Administration (NOAA) and based on the daily anomalies of $700\,\mathrm{hPa}$ geopotential height south of $20°$ S. In this study, we used the daily index over the 2000–2022 period (https://www.cpc.ncep. noaa.gov/products/precip/CWlink/daily_ao_index/aao/aao.shtml, last visited: February 2023).

## 2.3 Snowpack temperature

The vertical profile of temperature in the upper snowpack is estimated with the surface energy budget and thermal diffusion model called Minimal Firn Model (https://github.com/ghislainp/mfm, last visited: February 2023 Picard et al., 2009, 2012; Domine et al., 2019) inspired by the Minimal Snow Model (Essery, 2004). The model takes as input the $2\,\mathrm{m}$ air temperature, surface pressure, $10\,\mathrm{m}$ wind speed, $2\,\mathrm{m}$ specific humidity, surface downwelling long–wave radiation flux, surface downwelling



short–wave radiation flux, extracted from the ERA5 reanalysis. Besides, the required snowpack properties are taken from the literature: the density at the surface and at $10\,m$ depth (e.g. Leduc-Leballeur et al. (2015); Tian et al. (2018)), the thermal conductivity (Boone, 2002), the specific heat capacity of ice (Picard et al., 2009) and the temperature at $10\,m$ depth (equivalent to the mean annual air temperature in the Antarctic dry zones, Wang and Hou (2010)). From the MFM outputs, we used the temperature at $10\,cm$ depth and the temperature gradient in subsurface, defined as the difference between temperatures

estimated at 0 and $1\,cm$ in depth (positive gradient when the surface is colder than the subsurface).

## 3  Methods

To investigate the atmospheric processes which generate the highest snow grain size occurrences over the East Antarctic Plateau, we used the GSI defined by Picard et al. (2012) as a proxy for the snow grain size. We selected four case studies. In particular, we studied the GSI summer changes in order to compare the mean variations with the variation during which occurs

the highest GSI and we analysed the large-scale atmospheric circulation during each case study.

### 3.1  Detection of the highest GSI over East Antarctica

We define an extreme event of grain growth as GSI being higher than 0.23. This value is the 99.5th percentile of all the maximum GSI over the dry area in Antarctica in the period 2000–2022 and is significantly higher (*p-value* = 0.012) than the mean annual maximum values of $0.11\pm0.01$ typical of the East Antarctica ice divide. Fig. 1 shows the maximum GSI over

2000–2022 derived from AMSU-B observations. Four extreme events have been selected (symbols in Fig. 1): the first in 2001 at $81.06°$ S, $63.12°$ E, near the Pole of Inaccessibility (Rees et al. (2021), referred to *A2001* as hereafter); the second in 2007 at $79.64°$ S, $82.97°$ E, near Dome A (*B2007*); the third in 2016 at $76.60°$ S, $98.42°$ E, near Lake Vostok (*C2016*) and the fourth in 2020 at $77.38°$ S, $39.08°$ E, very close ($\sim15\,km$) to Dome Fuji (*D2020*).

### 3.2  Identification of the seasonal increases in GSI

The annual onset and ending dates of the increase in GSI have been determined by means of the algorithm *Bayesian Estimator of Abrupt change, Seasonal change, and Trend* proposed by Zhao et al. (2019). In this algorithm, the GSI time-series are decomposed by numerous alternative models, their relative usefulness is quantified and they are combined into a better model via Bayesian model averaging. Thus, an estimation of the probabilities of change-point occurrences are provided by identification of abrupt changes in the slope of the GSI time-series. The seasonal grain size increase (*SGSI* from now on) is defined as the

period which covers the interval from the onset to the ending dates.

We applied this algorithm on the 2020–2022 time-series at the four locations. Because of the GSI intrinsic variability and the presence of missing data, the onset and ending dates are not always identifiable. Thus, over a total of 88 summers (i.e. 22 years at four locations), both the onset and ending dates were identified in 55 cases (62.5%) and are used for our analysis.





### 3.3 Principal Component Analysis

Principal Component Analysis (PCA) is used to identify the main behaviour of variables involved in the SGSI onset and ending. It is performed by using the 55 SGSI identified in Section 3.2 with two configurations: in the first one, day 0 is the onset date and the analysis is performed between 10 days before and 20 days after day 0. In the second, day 0 is the ending date and analysis is performed from 20 days before to 10 days after day 0. PCA is applied to the daily time-series of GSI, skin temperature, 10 m wind speed, snowfall, surface pressure, cloud cover, ice temperature at 10 cm depth and the surface ice

temperature gradient between 0 and 1 cm in depth. In order to apply the PCA algorithm, we used the *FactoMineR* library in *R* (https://cran.r-project.org/web/packages/FactoMineR/index.html).

### 3.4 Statistics of co-occurrences

To assess whether the SGSI onset date is related to an atmospheric river occurrence or not, we selected 1000 random dates in the period between November and December (when the onset usually happens) and we calculated the frequency of the

atmospheric rivers (defined by either the IWV or the vIVT classification) in a 5-day interval centred on those dates. Hence, an atmospheric river is associated to an onset date if it occurs in the interval [-2;+2] days around that date. We tested if the atmospheric rivers interval follows a binomial distribution with both a probability $p$ equal to its frequency and a parameter $n$ equal to the number of onsets. From this, a *p-value* is obtained indicating if we can reject the null hypothesis that atmospheric rivers and onsets occur independently.

## 4 Results

### 4.1 SGSI characterization

Fig. 1 shows that, in East Antarctica, the highest GSI occurred on the ice divide where elevation is higher than 3000 m and the slope is less than $0.1°$ (at 1 km resolution) (Slater et al., 2018).

The GSI time series of the four extreme events are reported in Fig. 2. A clear seasonal oscillation is observed at the four

locations with the maximum occurring during summer, but with a large inter-annual variability.

The SGSI is characterized by an increase in GSI of $0.14\pm0.01$ on average with respect to the annual minimum, and it generally takes place between the end of November and the beginning of February. The following decrease in GSI tends to be slower and lasts until the next spring.

From 55 cases selected over 2000–2022 (cf. Section 3.2), we found that the SGSI lasts $54.9\pm2.4$ days on average, with a

minimum of 28 days and a maximum of 114 days. The mean Julian day for the onset date is $336.8\pm1.7$ (i.e. 3 December) and for the ending date is $25.6\pm2.0$ (i.e. 26 January). The growing rate, defined as the total increase in GSI during the SGSI divided by its duration (in days), on average is $0.0026\pm0.0001$ day$^{-1}$.

We analysed variations of skin temperature, surface temperature gradient, temperature at 10 cm depth, snowfall, cloud cover and wind speed at 10 m during the SGSIs in the locations of the four extreme events. We computed the anomaly of selected me-

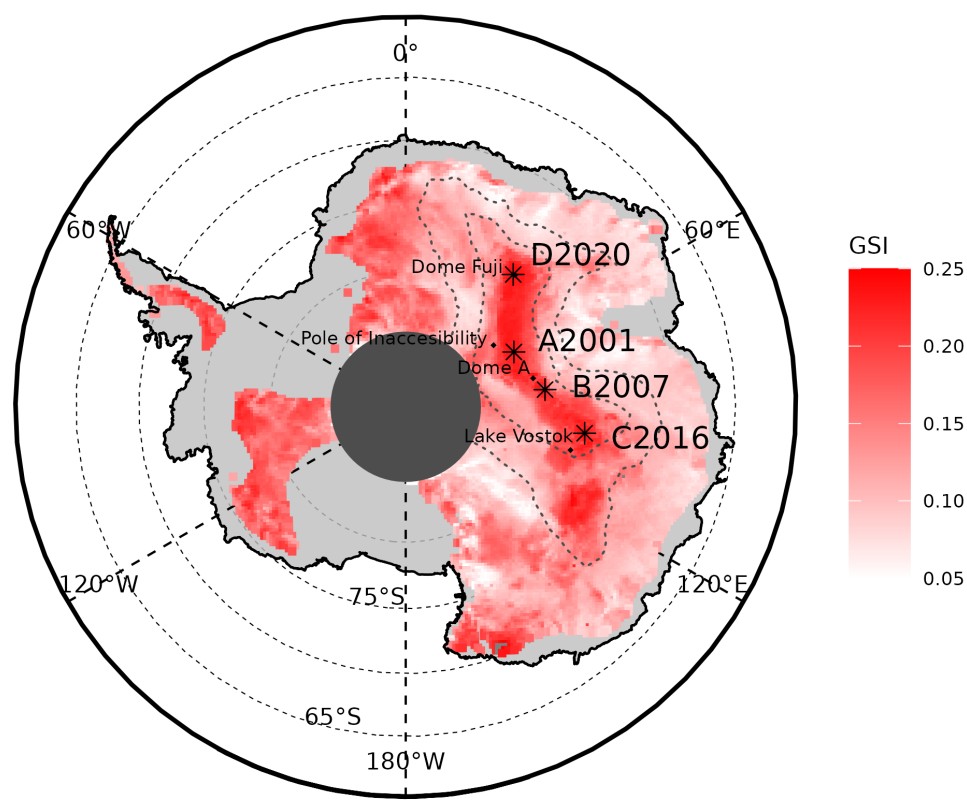

**Figure 1.** Map of GSI maximum for the 2000–2022 period (colours) with the elevation contours for 3000 m and 3500 m (dotted lines); the locations of the four extreme grain size events detected are marked with asterisks; the melting area is masked.

teorological variables with respect to the 2000–2022 period and we performed linear correlations with the seasonal maximum values of GSI. The correlation coefficients with the associated p-values are reported in Table 1. For the 55 selected SGSIs, a negative significant correlation is found with skin temperature (-0.41) and 10 m wind speed (-0.49) as well as a positive significant correlation with surface temperature gradient (0.39). Thus, wind and temperature appear to be the main drivers in determining the maximum GSI.

Some weaker correlation but significant at the 90% level, are obtained with temperature at 10 cm depth, snowfall and cloud cover (-0.28, -0.26 and -0.25, respectively), suggesting that these factors can also be involved in the maximum GSI. Conversely, no correlation is found for surface pressure.

Note that Picard et al. (2012) computed correlation between the increase in GSI during 1 December–15 January and accumulated precipitation during the same period at Dome C (75.10° S, 123.33° E) excluding the years with accumulation larger than 4 kg m$^{-2}$. Using the same approach, we obtained a correlation of -0.30 at our four locations over 2000–2022. They found a high correlation of -0.83 at Dome C over 1999–2010, but if we extend the calculation to the period 2000–2022, we obtain a

**Figure 2.** GSI time series for the four selected locations from January 2000 to May 2022: a) A2001; b) B2007; c) C2016; d) D2020. The 0.23 threshold is shown in red lines.

much weaker correlation of -0.44 at Dome C. We also are using the up-to-date ERA5 instead of ERA-Interim (Simmons et al., 2007).

Lastly, we studied the meteorological conditions during the SGSI onset and end by using the Principal Component Analysis
applied to the selected 55 cases, as described in Section 3.3. Fig. 3a shows the first components for the beginning of the SGSI. GSI starts to rise around day 0, i.e. the onset date. An increase in surface pressure and in skin temperature is observed with a peak 1-2 days after the onset. The temperature gradient over the $10\,\mathrm{cm}$ under the surface features a decrease a few days after





**Table 1.** Linear correlation coefficients between some meteorological parameters anomalies and the maximum GSI of the identified 55 SGSIs. Significance over 95% is grey and over 90% is pale grey.

|  | Correlation | p-value |
|---|---|---|
| Wind speed 10 m | -0.49 | 0.00016 |
| Skin temperature | -0.41 | 0.00210 |
| Snowfall | -0.26 | 0.05905 |
| Cloud cover | -0.25 | 0.06484 |
| Surface temperature gradient | 0.39 | 0.00366 |
| Temperature at 10 cm depth | -0.28 | 0.04129 |
| Surface pressure | -0.05 | 0.68960 |

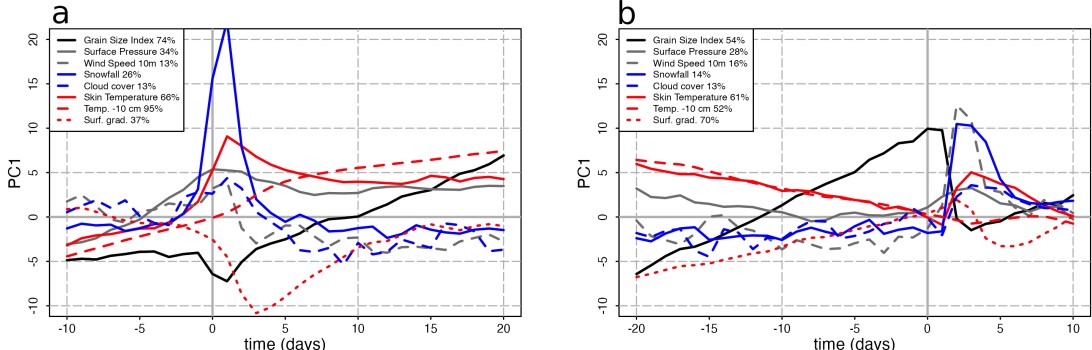

**Figure 3.** First components of the Principal Component Analysis performed over GSI, skin temperature, 10 m wind speed, snowfall, surface pressure, cloud cover, temperature at 10 cm depth and the surface temperature gradient between 0 and 1 cm in depth during a) the onset and b) the end of the SGSI. The proportion of explained variance for each parameter is reported in the legend.

the onset, with a minimum at day 3, and then it increases, while temperature at 10 cm depth keeps increasing for the whole range of 30 days considered, with the maximum rate between day 0 and day 5. Snowfall has a sharp peak during day 0 and day 1, and at the same time, 10 m wind speed and cloud cover rise. Then these three parameters decrease and exhibit lower stable values after day 5. The proportions of variance explained by the first component (i.e. for each parameter the amount of its variance the first component explains) of the GSI and temperatures at skin level and at 10 cm depth are high (74%, 66% and 95%, respectively), a bit lower for surface pressure and surface temperature gradient (34% and 37%) and even lower for wind speed, snowfall and cloud cover (13%, 26% and 13% respectively), probably because these parameters have a much higher natural day-to-day variability.




Fig. 3b reports the first components of PCA for the ending of the SGSI. GSI first component reaches its maximum at day 0, then sharply decreases at day 2. Note that during 2-3 days from day 2, all variables have change in behaviour: skin temperature and temperature at 10 cm depth showed a long term decrease, but then the first rises (during days 1-3) and the second stabilizes; the opposite variations are observed for surface temperature gradient, which exhibited a decreases during days 1-5. Snowfall, wind speed, surface pressure and cloud cover, which were low and relatively stable, have a sharp peak during days 2-3. This particular combination seems to determine the seasonal halt of the snow grain growth. The proportions of variance explained are quite similar to the onset case, with the exception of the surface temperature gradient which is higher while the temperature at 10 cm depth and the GSI which are lower, indicating a greater variability of these variables during the SGSI end.

### 4.2 The four most extreme maximum grain size events

During the selected four extreme events (see Section 3.1), GSI increased by 0.24–0.27 between the onset and end dates, and reached values above 0.23 (Table 2). Moreover, the GSI remained significantly above the climatological average (2000–2022) for the whole summers and subsequent autumns, with extreme standard deviations of over $+3\sigma$ during these maximums. The duration of these events ranges from 39 to 90 days.

In order to evaluate the spatial extent of the regions featuring high GSI events, we defined the extent of each of the 22 summer seasons as the area with the seasonal maximums GSI larger than 0.20, i.e. the 95th percentile of all the maximum GSI over the dry area in Antarctica in 2000–2022. On average, annually this area is $40\,000\pm10\,000\,\mathrm{km}^2$. The extent of the most extreme events ranges from $26\,000$ to $178\,000\,\mathrm{km}^2$. The magnitude, extent and growth rate of the A2001 and D2020 events are similar, while the B2007 and C2016 events feature a slightly larger maximum but smaller growth rate and spatial extent.

**Table 2.** Coordinates and elevation of the four extreme grain size events, and characterization of their SGSI.

| Event | Coordinates | Elevation | Extent | Max GSI | SGSI | Duration | Growing rate |
|---|---|---|---|---|---|---|---|
| | | m | km$^2$ | | | days | day$^{-1}$ |
| A2001 | 81.06° S, 63.12° E | 3891 | 178 000 | 0.234 | 01/12/2000–25/01/2001 | 55 | 0.0046 |
| B2007 | 79.64° S, 82.97° E | 3914 | 26 000 | 0.253 | 05/12/2006–05/03/2007 | 90 | 0.0030 |
| C2016 | 76.60° S, 98.42° E | 3669 | 46 000 | 0.244 | 14/12/2015–14/02/2016 | 62 | 0.0039 |
| D2020 | 77.38° S, 39.08° E | 3794 | 140 000 | 0.237 | 09/12/2019–17/01/2020 | 39 | 0.0060 |

Fig. 4 to 7 show GSI and some snow and atmospheric variables during the summer seasons of the most extreme events and their 2000–2022 climatology for comparison.

The A2001 event (Fig. 4) started on 1 December 2000 and the SGSI lasted until 25 January 2001. The GSI rate, defined as the daily variation averaged over a 10-day window, was maximum one week after the onset, when the sky was mainly clear. After the growth ending date, snowfalls became more frequent and the cloud cover and wind speed were often above the climatological mean. In February, the GSI rate was about zero and began to become negative later in March.



**Figure 4.** The A2001 event timeseries daily from 1 November 2000 to 31 March 2001 of a) GSI, and b) the GSI rate over a 10 day window; c) temperature at 2 m (dark blue), skin temperature (blue), ice temperature at 10 cm depth (pale blue), and d) ice temperature gradient between 0 and -1 cm (grey line); e) wind speed at 10 m; f) surface pressure; g) cloud cover; h) snowfall. The 2000–2022 climatology of each variable is in dotted line. The SGSI period is in grey area.

The B2007 SGSI (Fig. 5) lasted longer than the other extreme events, between 5 December 2006 and 5 March 2007. Despite the sudden decrease in GSI on 13 January coinciding with a large snowfall, GSI continues to increase after and the SGSI





lasted 90 days, which is much longer than the mean (54.9 days, cf ) and the other extreme events (short than 62 days). Note that the GSI rate remained positive but decreased in January when more frequent snowfalls and higher wind speed happened. This events differs from the other extreme events by a lower mean snowfall anomaly which during the SGSI was nearly zero

(-3% with respect to the 2000–2022 climatology), unlike the other extreme events which had very negative snowfall anomalies (equal or less than -50%). Besides, excluding the days immediately after the onset, the sky was generally cloudy during the whole SGSI (a non-significant +2% in total with respect to climatology during the SGSI, while the anomalies for the other extreme events were negative, from -7% to -23%).

The C2016 event (Fig. 6) started on 14 December 2015 and stopped on 14 February 2016. The GSI rate was steadily positive

until mid-January then began to decrease. The wind was generally below the climatological mean, and snowfalls were scarce. Starting from the end date, when a snowfall occurred, the wind speed suddenly rose above average, and remained steadily high.

Finally, the onset date of the D2020 event was on 9 December 2019 (Fig. 7), and GSI increased until 17 January 2020. The GSI rate was high (nearly +0.1/10 days) from the onset to the end of December, and the growing rate averaged over the SGSI was the highest with respect to the other extreme events (see Table 2). The low wind speed condition around the

onset was favoured by a high pressure situation, above the climatological mean by nearly $20\,\mathrm{hPa}$ in the days right before the onset (the highest value among the extreme events). Overall, calm wind condition was maintained for the rest of the SGSI, recording the largest negative anomaly ($-0.8\,\mathrm{m\,s^{-1}}$) among the extreme events. Actually, the onset occurred just after a large precipitation event observed over the 8–10 December period ($1.1\,\mathrm{kg\,m^{-2}}$ of snowfall), while the other extreme events had smaller accumulations around the onset. Note that snowfall of 8 and 9 December 2019 were both above $0.4\,\mathrm{kg\,m^{-2}}$,

corresponding to a precipitation event ranking above the 95th percentile for daily snow accumulation as estimated by Dittmann et al. (2016). Very few snowfall was recorded during the rest of the SGSI. The wind speed increased at the SGSI end, and in the following days wind speed above the climatology often occurred accompanied by snowfalls.

Table 3 summarises the anomalies in wind speed at $10\,\mathrm{m}$, temperatures, snowfall, cloud cover, surface temperature gradient and surface pressure during the SGSIs of the four most extreme grain size events. Even though these four cases had different

durations, some common features appear: the presence of snowfall precludes the onset accompanied by negative surface temperature gradients, a low wind speed situation during a few days following the onset, with also low temperatures, especially at skin level, and very few snowfalls and low cloud cover, except for the B2007 event (Fig. 5) which had frequent snowfall events. No clear conclusion can be drawn for the surface pressure anomaly, since two events occurred with positive mean anomalies and two events with negative ones. However, in the 10-15 days around the onset, the pressure anomaly was positive in all cases.

**4.3 On the role of atmospheric rivers**

The SGSI onset and end is generally associated with stormy conditions inducing large precipitation amounts, as shown in Fig. 3. Recently, Wille et al. (2021) demonstrated that atmospheric rivers (ARs), which facilitate the intrusion of relatively warm and moist air masses from the lower latitudes into Antarctica, generate extreme snowfall events. They also play a leading role in determining the Antarctic precipitation variability on inter-annual timescales, especially on the East Antarctica Plateau where

the snowfall accumulation is overall low. They estimated that 10–20% of the annual accumulation across East Antarctica





**Figure 5.** As in Fig. 4 but for the B2007 event, from 1 November 2006 to 31 March 2007.

is caused by atmospheric rivers. Along the highest East Antarctic ice divide, large amount of snow can be generated under blocking anticyclonic circulations or ridging situations which accompany the moist air going to the pole (Hirasawa et al., 2013; Pohl et al., 2021).

Here, we studied the possible link between the ARs presence and the SGSI. Assessing the co-occurrence of ARs and the onset date requires to compare the frequency of the atmospheric river around the onset date, and perform a statistical test





**Figure 6.** As in Fig. 4 but for the C2016 event, from 1 November 2015 to 31 March 2016.

to determine if this frequency is significantly different from that obtained with random dates (cf. Section 3.4). The mean frequencies of atmospheric river occurrence in 1000 random 5-day intervals centred between November and December in 2000–2022 are $p = 9.3\%, 6.8\%, 7.2\%, 8.9\%$ for the A2001, B2007, C2016, D2020 locations, respectively. $m = 5, 3, 5, 3$ onsets were associate with at least one atmospheric river in the 5-day window centered on each onset. Considering the number of

onsets available for each location are $n = 13, 17, 13, 12$, respectively and comparing to a binomial distribution with these



**Figure 7.** As in Fig. 4 but for the D2020 event, from 1 November 2019 to 31 March 2020.

$n$ and $p$, we obtained that the $m$ have *p-value* = 0.0006, 0.0249, 0.0002 and 0.0173 for location A2001, B2007, C2016 and D2020, respectively. It means that between 2000 and 2022 the onsets and the atmospheric rivers are very unlikely independent. Thus, we can reject the null hypothesis that atmospheric rivers and the onset of the grain SGSI are independent. However, it is important to note that not all the onsets are associated to an atmospheric river. On average, for the four locations, atmospheric rivers are associated with the onset in 38%, 18%, 38% and 25% of the cases. Additionally, no atmospheric river approached



**Table 3.** Mean anomalies with respect to the 2000–2022 climatology during the SGSIs of the most extreme events. Significance over 95% is grey.

| Event | Wind speed | Skin T | T 2 m | T -10 cm | Snowfall | Cloud cover | Surface T gradient | Surface pressure |
|---|---|---|---|---|---|---|---|---|
| A2001 | -0.7 m s$^{-1}$ | -1.1 K | -1.0 K | -0.6 K | -50% | -23% | +0.3 K/m | +2.2 hPa |
| B2007 | -0.7 m s$^{-1}$ | -1.8 K | -1.4 K | -1.3 K | -3% | +2% | +0.7 K/m | -0.9 hPa |
| C2016 | -0.6 m s$^{-1}$ | -2.1 K | -1.7 K | -1.5 K | -60% | -7% | +0.8 K/m | -3.5 hPa |
| D2020 | -0.8 m s$^{-1}$ | -1.2 K | -0.5 K | -0.4 K | -51% | -23% | +0.5 K/m | +1.9 hPa |
| Average | -0.7 m s$^{-1}$ | -1.6 K | -1.2 K | -1.0 K | -41% | -13% | +0.6 K/m | -0.1 hPa |

the four locations during the onset of the most extreme SGSIs investigated in Section 4.2. Therefore, atmospheric rivers are sometimes involved in the onset but they are not a required factor and they do not play a role in determining the maximum seasonal grain size. A similar analysis was conducted also to assess the co-occurrence of ARs and the end date of the SGSIs and results were quite similar. We obtained *p-value* = 0.0360, 0.0013 and 0.0001 for A2001, B2007 and D2020, respectively, and a non-significant 0.1444 for C2016. Hence, at least in 3 of the 4 locations, there is a significant connection between ARs and the ending of the SGSI, with the 15%, 24%, 8% and 42% of the ending dates associated to an AR, respectively.

## 5 Discussion

### 5.1 Grain growth

Our analysis shows that the amplitude of the summer increase of GSI is highly variable and large anomalies emerge occasionally during the summer, giving rise to very high values of grain size, localised in specific areas. We found some interactions between grain size evolution and local meteorological conditions at different time scales, from daily to seasonal, along the main East Antarctic ice divide. Wind and skin temperature have been shown to play a leading role in affecting snow metamorphism during the austral summer with correlation of -0.49 and -0.41, respectively. We observed that the beginning of the growth usually takes place just after a snowfall: at first, the grain size decreases during the precipitation, as new smaller grains are brought onto the surface (Domine et al., 2007). However, the snowfall is usually associated to a large near-surface temperature gradient which is a driver of the dry snow metamorphism causing a rapid increase of the grain size (e.g. Colbeck (1982), Grenfell et al. (1994)).

The East Antarctic Plateau cannot be regarded as thermally homogeneous (González et al., 2021). For example, cold air is transported downhill from the domes and the ice divides triggering katabatic winds, and shallow depressions exist on the flank of the ice divides where pooling of drained cold air allows greater radiative heat loss and hence enhanced cooling of the surface (Scambos et al., 2018). Besides, because of this top position, the ice divides are characterised by a general wind divergence





(Parish and Bromwich, 2007) and weak wind conditions (Yamanouchi et al., 2003). In addition to the low accumulations, these features are favourable to the presence of large snow grains since high-speed winds tend to transport and deposit small ice particles on the surface, burying larger grains present. Moreover, lower than normal skin temperature observed during the

SGSIs of the most extreme events, suggests that a large temperature gradient is possible in the first centimeters of the snowpack and hence an enhanced flux of water vapour could appear from the subsurface to the surface, leading to the formation of recrystallization crystals (Flanner and Zender, 2006; Champollion et al., 2013; Leduc-Leballeur et al., 2017). Cold surface conditions could result from the generally clearer than normal sky conditions observed during these phases, favorable to local intense radiative cooling of the surface and an enhanced solid-state greenhouse effect (Dombrovsky et al., 2019), causing a

strong thermal gradient in the uppermost part of the snowpack, as observed in Gallet et al. (2014). The solid-state greenhouse effect could not be appreciated in this study as we used the python version of the MFM, in which its parametrization is absent. The low wind speed also hinders the dispersion into the atmosphere of the sublimated snow favouring local supersaturation conditions and, hence, the growing of grains.

## 5.2   Link with large scale mean conditions

As demonstrated by Wille et al. (2021), the zonal distribution of the $500\,\mathrm{hPa}$ geopotential height anomalies associated with ARs landfalls resembles the negative phase of the SAM. We thus analysed if a possible connection between GSI increase and large scale atmospheric circulation also exists. The $500\,\mathrm{hPa}$ geopotential height can help identify large scale atmospheric structures, such as the tropospheric polar vortex (Kwon et al., 2020; Gordon et al., 2022). Fig. 8 shows the ERA5 $500\,\mathrm{hPa}$ geopotential height mean standardized anomalies (with respect to 2000–2022 climatology) for the 5-day intervals centered on

the available onset dates for each location. There are mostly positive anomalies all over the continent, with significant values in the nearby of the considered locations, with waviness in the Southern Ocean. For the four study cases, we also observe a positive anomaly in the Amundsen Sea, possibly related to a low activity of the Amundsen Sea Low (see for example Turner et al. (2012)).

Some tests were performed in order to assess a more frequent occurrence of particular zonal wave numbers during the 55

onsets but the results were not conclusive. Transient waviness (wavenumber 3 or 4) can be seen in Fig. 8 even though not at significant level, hence future work will study in more detail the possible involving of mid-latitude atmospheric dynamics and zonal wave activity.

As a matter of fact, studying the Antarctic Oscillation index we found that the period October—November—December 2019 (the months before the D2020 event) recorded the lowest mean value of the AAO index since the beginning of the series in

1979 (Lim et al., 2021). This was the result of a minor sudden stratospheric warming (SSW) that occurred in September 2019 and propagated downwards, thereby causing anomalous tropospheric circulation from the end of October to mid-December 2019 (Shen et al., 2020). Such stratospheric warming events are very rare in the Southern Hemisphere, as their occurrence has been estimated to 4% per year (Wang et al., 2020). However, this did not happen before the other three extreme snow grain size events. Hence, the average AAO around the onset cases considered in this study is not statistically different from any other date

during November-December. During the four most extreme events most of the time (three cases out of four) a negative AAO





index is present, i.e. a weak polar vortex situation. In particular during the 5-day intervals centred on the onsets of the four most extreme events we have an average of the AAO index of -1.48 (A2001), 0.18 (B2007), -1.71 (C2016) and -2.09 (D2020). A negative AAO situation seems to appear more preferential to trigger SGSI. However, a negative AAO is neither necessary nor is sufficient. Not necessary as other onsets had very negative AAO values (not shown) and they did not ended in high GSI

later in the season. Not sufficient as for example the AAO for the B2007 event is slightly positive. In fact, over the 55 selected cases, 31 had negative AAO mean values in the 5-day intervals centred on the onsets, with an average of -0.39±0.19 over all the cases, with a *p-value* of 0.078 on a two-tailed t-test. Thus, a negative AAO index is more present than the positive phase during the onsets even though this does not reach the 95% significance level.

## 6   Conclusions

We studied the seasonal variations of the grain size in East Antarctica using an index computed using the 150 and 89 GHz observations from the satellite instrument AMSU-B, and previously introduced and analysed at Dome C (Picard et al., 2012). The present study covers the interval 2000–2022, the entire period of availability of the remote-sensed snow grain size index from AMSU-B sensors, over which we identified 55 seasonal grain size increases (SGSI) in four locations with identifiable onset and end dates. Using the ERA5 reanalysis, we established that grain size signal is linked to particular atmospheric

conditions. In order to study these favourable circumstances, we identified four events of extreme increase of the grain size, in different locations along the highest ice divide, and compared them to the usual SGSIs between 2000 and 2022. We searched for connections with local meteorological conditions and then with the synoptic atmospheric background.

The growth onset of the snow grains is usually linked to a snowfall event, in one third of the considered 55 cases associated with the presence of an atmospheric river providing warm and moist air into the inner plateau. The final maximum value of the

grain size, that is usually reached in January-February, is determined by what happens in the weeks following the onset: the common features during the SGSI which emerge from this study are low wind speed conditions and skin temperature below the climatological mean.

We observed that ARs and, to a lesser extent, weak polar vortex conditions were related to the SGSI onset and end. However these situations are not necessary conditions. Another potential connection may exist with the Amundsen Sea Low, but our

study did not allow to accurately characterize if a co-occurrence of different critical conditions could provoke a large GSI event. Further analysis could be done using the ERA5 reanalysis in order to extend the time interval and evaluate the significance of these anomalous situations and possibly to discover some other similar configurations which had led to intense growth of the snow grain. The microwave data available in the past (e.g. Special Sensor Microwave Imager, SSM/I) miss the 150 GHz channel, but the 85–91 GHz channel could be used to find abrupt decreases in 1987–1999. However, the extension is not trivial

as the grain size index is indirect and results from remote sensed observations. In situ observations should be included in the analysis, when available, in order to correct possible biases.

This analysis allowed improving the understanding of the causes of the SGSI onset and end, andcould be important for projections of the frequency of extreme GSI events in a changing climate.



**Figure 8.** Mean standardized anomalies of the 500 hPa geopotential height fields from ERA5 over the period between 2 days before and 2 days after each synchronised onset date available for each location with respect to the 2000–2022 climatology. The number of available onset dates are 13, 17, 13 and 12, respectively. Brighter colours highlight the value with significance over 90th on a two tailed t-test. Asterisks mark the locations: a) A2001 b) B2007 c) C2016 d) D2020.

*Data availability.* ERA5 reanalysis data are available from https://cds.climate.copernicus.eu/#!/search?text=ERA5&type=dataset (Accessed
on 05-Aug-2022). The AMSU-B $T_B$ values were retrieved at https://www.ncei.noaa.gov/access/metadata/landing-page/bin/iso?id=gov.noaa.



ncdc:C00981 (NCEI DSI 3702_01 dataset). AAO index were retrieved from https://www.cpc.ncep.noaa.gov/products/precip/CWlink/daily_
ao_index/aao/aao.shtml.

*Author contributions.* CS led the study and performed the analysis, GM, MLL, GP, VF and BP supervisioned the discussion. MLL and GP
helped provide the brightness temperature data and VF provided the atmospheric river datasets. All authors contributed to revise manuscript.

*Competing interests.* The authors declare that they have no conflict of interest.

*Acknowledgements.* Hersbach et al. (2018a, b) was downloaded from the Copernicus Climate Change Service (C3S) Climate Data Store.Vincent
Favier acknowledge the support from Agence Nationale de la Recherche, project ANR-20-CE01-0013 (ARCA).



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
