# Peer review of "Extreme events of snow grain size increase in East Antarctica and their relationship with meteorological conditions"

_The Cryosphere, 2023_

## Referee Comment (RC2)

**Review: "Extremes of surface snow grains change in East Antarctica and their relationship with meteorological conditions"**

Stefanini et. al.

For publishing in The Cryosphere

**1 General**

This paper reviews the conditions causing extreme changes in surface grain size in east Antarctica. I like how this paper uses case studies over different geographic area to expand upon previous research, and the choice of location seems very logical. I think this paper could use a little more clarity regarding the link of "large scale mean conditions", as although the link to precipitation, wind speed, and temperature is very well established the link to larger scale climatology in section 5.2 needs more work. Links to figures/tables are strong and this paper overall establishes a solid connection of common features to increase in grain size in this region.

**1.1 Major Comments**

- **Antarctic Oscillation Index** The link to antarctic oscillation index is not well explained: was there any expected link to AAO? Why look into this parameter?

- **Atmospheric Rivers:** The link to AR's and grain growth needs to be more clear in the abstract that it is due to precipitation variability that there is thus grain growth. Although it specifies this later in the paper, the occurrence of warm moist air can lead to precipitation which can lead to grain growth, not just warm moist air can lead to grain growth (or else this would not a dry metamorphism in the snow pack.)

- **title** This paper is about the extreme large grain sizes, not both extremes and this might be good to represent in the title.

**2 Line Comments**

- Line 6: pressure blocking/ridge situation could probably just say pressure blocking and explain ridge situation in AR section.

- Line 112: What causes missing data? what causes dates to not be identifiable? Only being able to identify dates in 62.5% of cases, please comment on what bias this might potentially have.

- Figure 2: instead of saying 0.23 threshold maybe include "extreme growth threshold"

- Line 199: "this events" to "these events"

- Line 216: "Very few snowfall", wording, "very little snowfall"

- Line 332: "andcould" to "and could"

---

## Author Comment (AC1)

**Extremes of surface snow grains change in East Antarctica and their relationship with meteorological conditions**

**Stefanini et al., 2023**

**Anonymous Referee #1**

This paper reports on the temporal and spatial occurrence of extreme snow grain sizes on the interior plateau of East Antarctica, as observed by satellite. Four events are discussed in more detail, but explanations in terms of climatological forcing remain inconclusive. This is an interesting paper; the results are original but the analysis lacks depth and remains descriptive, reducing the impact. The figures are of good quality and the clarity of the writing is sufficient.

Thanks to devote time to the review of our manuscript. The manuscript has been revised, according to comments and suggestions provided. In particular we improved the discussion related to the relationships with climatological forcing and the physical processes involved in the GSI changes. Overall, we show here that the relationship with AR is robust independently of the reanalyses used and that ARs are sometimes involved in the onset, but they are not a necessary condition, and they do not play a role in determining the maximum seasonal grain size. We also extended our discussion on the large scale circulation impact with consideration of events that did not fit with ARs occurrences. Concerning the highly asymmetric seasonal cycles in the GSI shown in Figure 2, Picard et al. 2012 previously analysed these features, and we remind now in the manuscript the key role of temperature and of the subsequent increase of water vapor available transported through the snowpack, triggering dry snow metamorphism and hence a rapid increase of the grain size. Detailed responses on these two points are discussed hereafter.

Major comments

Title: the title is not clear, please find an alternative (e.g., Extreme changes in East Antarctic snow grain size and their ...)

Considering a similar suggestion from Referee #2, the new title is "Extreme events of snow grain size increase in East Antarctica and their relationship with meteorological conditions".

Section 2.2.2: Even though AR detection is only weakly sensitive to the re-analysis product used, it is still puzzling why not the same re-analyses were used for the AR detection and meteorological analysis. Please comment. A major concern is the accuracy of ERA5/MERRA over this extreme environment in interior East Antarctica, yet these data form the basis of much of the discussions. Please comment.

We used MERRA-2 ARs catalog since it was the one at our disposal when we did the analysis and considering that the AR detection frequency poorly depends on the ERA5 ot MERRA catalouge's selections as shown in Wille et al. 2021 (https://doi.org/10.1029/2020JD033788). Nonetheless, as the reviewer correctly pointed out and to be more coherent with the other analyses, outcomes from ERA5 have been added to the manuscript.

[revised manuscript text omitted]

Section 2.3: How can the subsurface temperatures be determined using daily average climate forcing (I. 65)? The daily cycle is considerable and important for grain growth, given its nonlinear dependence on temperature.

Thanks for your comment, hourly data are actually used as input of MFM but this was not specified in the text. It has been added at the end of Section 2.2.1: '*We used daily means from 2000 to 2022 and also hourly values as inputs for the thermal model*' and specified in the Section 2.3: '*extracted from the hourly ERA5 reanalysis*'.

I. 97: As the definition of an extreme growth event, I expect a growth rate, not an occurrence.

Since we are focusing on extreme size of snow grains, not growth rate, we propose to reformulate as: *We define an extreme grain size as GSI being higher than 0.23.*

I. 134: The GSI time series show a highly asymmetric seasonal cycle, it almost appears as if the signal is reset each year. It is therefore not surprising that correlations with climate variables are weak. This deserves a brief discussion at this location: how can this be understood from physical processes? How can we be sure that this is not an artifact of the used method and/or satellite data? The same holds for the connection between grain size and atmospheric rivers: how would they be expected to be physically connected?

The highly asymmetric seasonal cycles in the GSI shown in Fig. 2 have been previously analysed in Picard et al. 2012. They showed with model simulations that this asymmetric cycle observed by satellite observations is real and not dependent from the location. The seasonal increase in GSI is faster than the decrease because they are driven by very different mechanisms. The increase is rapid due to a positive feedback loop, as explained in Picard et al. 2012: when grain size increases in summer snow albedo decreases and thus absorption of solar radiation increases. Temperatures and temperature gradient increase, which in turn promotes metamorphism and grain growth below the surface. By modelling, they artificially disabled this feedback, i.e. the albedo is kept constant at its value of the beginning of the season, and observed that the growth rate over the season is reduced by about 20%. On the

other hand, the seasonal decrease in GSI is due to the slow burial of the summer layer by precipitation, which are very small in this region of Antarctica. Nonetheless, the amplitude of this seasonal cycle depends on years. Indeed, the yearly reset does not fully happen as the seasonal GSI minimum is on average -0.045, -0.031, -0.041, -0.049 respectively for the four locations with large standard deviations of 0.019, 0.017, 0.016, 0.022. Moreover, for example for A2001, we observe a period of constant increase of the minimum from 2008 to 2014, for reasons we did not explore.

In this study, we are interested in the seasonal increase, which is mainly driven by snow temperature, and temperature vertical gradient in the upper snowpack, and so by a complex combination of near surface meteorological variables. We added this part to Section 4.1: '*Seasonal increase in GSI is mainly driven by temperature and the subsequent increase of water vapor available to be transported within the snowpack upper layers (0–20 cm), triggering dry snow gradient metamorphism (Colbeck 1982). Summer precipitation contributes to inter-annual variability through the deposition of a thin layer of fine grains at the surface, which increases albedo and reduces the penetration of solar radiation and the heating of the topmost layer, thereby reducing the grain growth (Picard et al. 2012). In the same way, the snow transport by wind results in a sorting of the grains by size in the topmost layers, as smaller snow grains are expected to fall out last (Grenfell et al. 1994, Domine et al. 2007). By contrast, the seasonal increase of the grains is favored by high subsurface temperature vertical gradients (Colbeck 1993), mainly because of the solid-state greenhouse effect (Dombrovsky et al. 2019): the snow cools at the surface by emitting infrared radiation and warms a few centimeters below by absorbing solar radiation. This effect is particularly enhanced when the sky is clear. For these reasons, we analysed variations of skin temperature, surface temperature gradient, temperature at 10 cm depth, snowfall, cloud cover and wind speed at 10 m during the SGSIs in the locations of the four extreme events.*'. The relationship between GSI increase and ARs is related to the fact that in some cases snowfalls events are observed at the onset and/or at the end of the SGSI and these often occurred in presence of AR. This has been added in the text at the end of the same section, after the PCA analysis: '*Note that the snowfall peaks observed both at the onset and at the end of the SGSI suggest relatively strong precipitation events, which could be related to the occurrence of an AR. We investigate this in Section 4.3.*'.

l. 171 and following paragraph: The discussion remains very descriptive and potential correlations are discussed mainly in terms of variance explained; it would be more informative to identify physical processes that cause rapid grain growth, and then try to couple these to co-variations with meteorological parameters. See also comment on the seasonal cycle above.

The justification on the choice of these parameters has been added as explained in the previous comment.

Minor and textual comments

l. 18: insulation -> isolation

Corrected.

l. 21: tropics -> subtropics or mid-latitudes

Corrected.

l. 22: Consider moving this motivation to study grain size to the beginning of the introduction.

Thank for your comment. We moved this sentence at the beginning as you suggest it to help the motivation understanding.

l. 39: "high grain size" sounds awkward; suggest finding an alternative.

Corrected with "large grain size".

l. 65: suggest: "and we reprojected them using the Southern polar stereographic projection"

Corrected.

l. 65: mean -> means

Corrected.

l. 107: Please specify what you mean by 'models' here.

It has been better specified: *'the time-series are decomposed by numerous alternative statistical models which separate the seasonal, trend and residual contributions. Their relative usefulness is quantified and they are combined into a better model via Bayesian model averaging.'*

Fig. 2: axes labels are missing

Added.

l. 198: short -> shorter

Corrected.

l. 232: to the pole -> to the interior

Corrected.

l. 263: the downhill transport of cold air is a katabatic wind

Yes, we have rewritten as: *'katabatic winds transport cold air downhill from the domes and the ice divides, …'.*

**Anonymous Referee #2**

This paper reviews the conditions causing extreme changes in surface grain size in east Antarctica. I like how this paper uses case studies over different geographic area to expand upon previous research, and the choice of location seems very logical. I think this paper could use a little more clarity regarding the link of "large scale mean conditions", as although the link to precipitation, wind speed, and temperature is very well established the link to larger scale climatology in section 5.2 needs more work. Links to figures/tables are strong and this paper overall establishes a solid connection of common features to increase in grain size in this region.

Thanks to devote time to the review of our manuscript. The manuscript has been revised, according to comments and suggestions provided. Regarding the large scale climatology, we deeply considered this comment which was also related to the comment from Referee #1 concerning the role of ARs. To summarize, we observed that the only large scale features that clearly present a significant statistical link with GSI onset was AR occurrences. Thus, we analysed the role of these events more in details. We clearly observed that ARs are sometimes involved in the onset, but they are not a necessary condition, and they do not play a role in determining the maximum seasonal grain size. As ARs frequency in Antarctica have been previously linked in some regions with the SAM (Wille et al. 2021), we more deeply analysed the role of the SAM-like pattern on the onset of the grain size increase and on the evolution of each event. Moreover, we investigated the zonal wave number based on the geopotential height anomalies, as well as the convection anomalies in the tropics and the pressure anomalies in the stratosphere. However, no clear evidence of how the large scale circulation act on GSI has been demonstrated, and possible connection are not fully understood at the stage of the study.

Thus, we extended our discussion on the large scale circulation impact at the end of Section 5: '*In the Dome Fuji area during the anticyclonic circulation, a peculiar situation of warm-core eddy emerges near the top of the dome, while cooler air accumulates on the saddle on the eastern side, as described in Section 5.1. Indeed, on 5 December 2019, ERA5 reanalysis recorded the maximum gradient of about 4 K between the warm-core eddy and the saddle. Afterwards, temperature dropped until the beginning of January 2020, with an average anomaly of -4.3 K from 21 December 2019 to 4 January 2020, also observed in Fig. 7. The anticyclonic ridge developed when an intense wet oceanic air flow arrived over the ice sheet plateau through the Dronning Maud Land on 7–9 December and reached Dome Fuji, which became the center of the anticyclonic counterclockwise rotation of the flow. This event is not an AR according to the criteria defined in Wille et al. (2021), and neither the A2001 and B2007 events, when moisture intrusions arrived from Enderby Land and Amery Ice Shelf, respectively. Only the C2016 event is associated to an AR occurrence, which arrived from Victoria Land, on the basis of the ERA5 catalog, but not of the MERRA-2 catalog. Nonetheless, intense moisture advections were present during all these extreme events.*'

Major Comments

The link to antarctic oscillation index is not well explained: was there any expected link to AAO? Why look into this parameter?

We noticed that the general circulation at the onset of the four locations presents evidence of pressure difference between continent and ocean which are characteristic of SAM. We expanded our discussion on the SAM as following:

*'in Fig. 8 we can observe, in all the four cases, that there are some pressure differences between the continent and the ocean that are clearly reminiscent of the SAM. Thus, these seasonal events of grain size increases tend to correspond to negative SAM-like patterns.'* Similarly, in `https://doi.org/10.1029/2020JD033788` the authors show that the zonal distribution of the 500 hPa geopotential height anomalies associated with AR landfalls resembles the SAM and in a similar way, Fig. 8 shows the signature of AR landfalls.

The link to AR's and grain growth needs to be more clear in the abstract that it is due to precipitation variability that there is thus grain growth. Although it specifies this later in the paper, the occurrence of warm moist air can lead to precipitation which can lead to grain growth, not just warm moist air can lead to grain growth (or else this would not a dry metamorphism in the snow pack.)

Yes, we better specified it in the abstract as: *'[...] In these cases, the ERA5 reanalysis revealed a high pressure blocking close to the onsets of the SGSI. It channels moisture intrusions from the mid-latitudes, through atmospheric rivers that cause major snowfall events over the plateau.'*

This paper is about the extreme large grain sizes, not both extremes and this might be good to represent in the title.

As replied to a similar comment from the first reviewer, the new title is "Extreme events of snow grain size increase in East Antarctica and their relationship with meteorological conditions".

Line Comments

Line 6: pressure blocking/ridge situation could probably just say pressure blocking and explain ridge situation in AR section.

Corrected.

Line 112: What causes missing data? what causes dates to not be identifiable? Only being able to identify dates in 62.5% of cases, please comment on what bias this might potentially have.

Even in case all daily satellite observations are available when the GSI started to increase, the algorithm fails to provide an unambiguous date for the onset and end because of the large uncertainty provided by the statistical time series decomposition models in some location and year. Nevertheless, for each location this happens 5–10 time and more than half of the seasonal increase is well defined (13, 17, 13, 12, respectively; see Section 4.3) and uniformly distributed in the 2000–2022 period. None of the excluded SGSI led to unusual grain size index values. Even if the onset and end dates are not clearly identified by the algorithm, the seasonal increase of the GSI happens. Besides, we decided to perform the PCA analysis with the 55 univocal case in order to make results more reliable since we focus on the onset and end dates. Therefore, we expect that the use of only 62.5% of the total cases has a weak impact on our results.

Figure 2: instead of saying 0.23 threshold maybe include "extreme growth threshold"

Corrected.

Line 199: "this events" to "these events"

    Corrected.

Line 216: "Very few snowfall", wording, "very little snowfall"

    Corrected.

Line 332: "andcould" to "and could"

    Corrected.

---

## Referee Report (RR1)

**Final Review: "Extreme events of snow grain size increase in East Antarctica and their relationship with meteorological conditions"**

Stefanini et. al.

For publishing in The Cryosphere

**1 General**

Thank you for answering all questions and responding to all suggestions. Apologies for the long-awaited review back. All changes are positive and contribute to clear understanding in this paper. I think the title is an excellent choice, and using the term 'grain size increase' will help in literature review searching. Alongside with suggestions from referee #1 about large scale circulation impact, the extension contributes to reader understanding. The expanded discussion on SAM is satisfactory. Finally, this is a wonderful paper, and I look forward to sending it to colleagues.

**1.1 Line Comments**

- Figure 4/5/6: Cloud Cover units (%) ?

- Figure 4/5/6: GSI rate (per day?), add units.